# Dynamic Profile of the Yak Mammary Transcriptome during the Lactation Cycle

**DOI:** 10.3390/ani13101710

**Published:** 2023-05-22

**Authors:** Wei Xia, Yili Liu, Juan J. Loor, Massimo Bionaz, Mingfeng Jiang

**Affiliations:** 1College of Animal and Veterinary Science, Southwest Minzu University, Chengdu 610041, China; 2College of Animal Science and Technology, Hebei Agricultural University, Baoding 071000, China; 3Department of Animal Sciences, Division of Nutritional Sciences, University of Illinois, Urbana, IL 61801, USA; 4Department of Animal and Rangeland Sciences, Oregon State University, 112 Withycombe Hall, Corvallis, OR 97331, USA

**Keywords:** yak, mammary gland, transcriptome, lactation, dynamic impact approach

## Abstract

**Simple Summary:**

Although the transcriptome of dairy cow mammary tissue has been reported, the transcriptome of yak mammary tissue remains unknown. The objective of this study was to assess the transcriptome of the mammary tissue of four yaks during the whole lactation cycle. The statistical analysis identified >6000 differentially expressed genes (DEGs) throughout lactation, with a large number of DEGs observed at the onset (1 d vs. −15 d) and at the end of lactation (240 d vs. 180 d). Bioinformatics analysis revealed a major role of lactation-associated genes on BTA3, BTA4, BTA6, BTA9, BTA14, and BTA28. Functions affected by the transcriptomic adaptation to lactation in mammary tissue of yak were very similar to those observed in dairy cows. Our study provides fundamental resources on yak mammary tissue transcriptome for the research community.

**Abstract:**

The objective of this study was to assess the transcriptome of the mammary tissue of four yaks during the whole lactation cycle. For this purpose, biopsies of the mammary gland were performed at −30, −15, 1, 15, 30, 60, 120, 180, and 240 days relative to parturition (d). The transcriptome analysis was performed using a commercial bovine microarray platform and the results were analyzed using several bioinformatic tools. The statistical analysis using an overall false discovery rate ≤ 0.05 for the effect of whole lactation and *p* < 0.05 for each comparison identified >6000 differentially expressed genes (DEGs) throughout lactation, with a large number of DEGs observed at the onset (1 d vs. −15 d) and at the end of lactation (240 d vs. 180 d). Bioinformatics analysis revealed a major role of genes associated with BTA3, BTA4, BTA6, BTA9, BTA14, and BTA28 in lactation. Functional analysis of DEG underlined an overall induction of lipid metabolism, suggesting an increase in triglycerides synthesis, likely regulated by PPAR signaling. The same analysis revealed an induction of amino acid metabolism and secretion of protein, with a concomitant decrease in proteasome, indicating a major role of amino acid handling and reduced protein degradation in the synthesis and secretion of milk proteins. Glycan biosynthesis was induced for both N-glycan and O-glycan, suggesting increased glycan content in the milk. The cell cycle and immune response, especially antigen processing and presentation, were strongly inhibited during lactation, suggesting that morphological changes are minimized during lactation, while the mammary gland prevents immune hyper-response. Transcripts associated with response to radiation and low oxygen were enriched in the down-regulated DEG affected by the stage of lactation. Except for this last finding, the functions affected by the transcriptomic adaptation to lactation in mammary tissue of yak are very similar to those observed in dairy cows.

## 1. Introduction

The yak (*Bos grunniens*) lives throughout the Qinghai-Tibetan plateau of China. It is at home at high altitudes where a harsh climate exists, including low temperature, humidity, and oxygen levels, as well as strong winds, high ultraviolet radiation, and other hazards. The yak is crucial in this area, as it is the main source of meat and milk to the 6.5 million Tibetan people living in the Qinghai-Tibetan plateau area [1]. Unlike dairy cows, which can produce more than 3000 kg milk in a single lactation, the milk yield of the yak is extremely low (<500 kg in 180 d of lactation cycle). However, despite the low milk yield, yak milk contains higher concentrations of milk solids, protein, fat, organic calcium, and conjugated linoleic acid than Holstein cow milk [2].

The mammary gland is specialized in the synthesis and secretion of milk, and undergoes drastic structural and metabolic changes during the transition from pregnancy into lactation and from lactation to the dry period [3,4,5]. As observed in the cow, mouse, and human, these events appear closely driven by changes in the transcriptome. Although basic functions of the mammary gland have been previously described [6,7], understanding the complexity of the adaptations to lactation is an active area of research [8]. High-throughput tools, including microarray and RNA-seq, have made it possible to uncover changes in the transcriptome of the mammary tissue in several species [3,4,5,9,10]. Although RNA-seq has gained tremendous popularity [11], researchers continue to use microarrays as well, especially in studies with large sample sizes [12]. Moreover, microarrays are more popular in clinical settings due to easier data processing [13].

The biology of the mammary gland of the yak has received relatively little attention, although epithelial cells isolated from the yak mammary gland have been characterized [14]. These cells have been used to identify the proteins involved in mammary epithelial differentiation in this species [15]. However, transcriptome analysis, especially if performed through the whole lactation cycle, can provide further insights into the biology of the mammary gland of the yak. Prior work from our group using RTqPCR revealed a role of the transcription factor MYC on the adaptation of the mammary gland of the yak to lactation [16], and characterized the expression of protein synthesis-related transcripts during the whole lactation [17]. A recent study underlined the importance of non-coding RNA in the mammary gland of the yak during lactation and the dry period [18]. Our group performed metabolomics profiling of yak mammary tissue during the whole lactation, revealing changes in glucose and amino acid metabolism [14,19].

All of the above studies have provided important insights into the adaptation of the yak mammary gland to lactation; however, they do not provide a comprehensive picture. Due to the important insights provided by high-throughput analysis of the transcriptome associated with bioinformatic tools in the adaptation to lactation of the mammary gland in other species [4,5,20,21], it is important to study the whole transcriptome of protein-coding transcripts in yak mammary tissue using similar approaches. Therefore, in the present study we performed a transcriptome profile of yak mammary tissue during the whole lactation cycle and compared it with prior data obtained in dairy cows.

## 2. Material and Methods

### 2.1. Milk Yield Measurement

As previously reported [17], milk yield was measured just after calving and at 20, 40, 60, 80, 100,120,140, 160, 180, and 200 d relative to parturition. The milk yield values were the amount of milk collected twice a day (morning and night milking), and were adjusted for the milk consumed by the calves. It is estimated that yak calves consume approximately half the milk produced. We measured milk production in an additional ten yaks at the end of June. During the entire lactating period, at least 18 individual measurements were recorded for each yak.

### 2.2. Animals, Experimental Design, and Diet

All experiments performed in the present study were approved by Southwest Minzu University Institutional Animal Care Committee (permit number: 2012-3-1). Four multiparous MaiWa yaks from Southwest Minzu University Tibetan Research Center weighing 220–260 kg were selected. As is typical of the Tibet plateau, the yaks were grazed without any supplementation. All selected yaks calved at the end of June.

### 2.3. Mammary Gland Collection and Preparation

The mammary tissue was collected via biopsy, alternating between the right or left rear quarter of the mammary gland at −30, −15, 1, 15, 30, 60, 120, 180, and 240 d relative to parturition [3,22]. Biopsy of the mammary gland was performed by a veterinarian. Briefly, to obtain mammary parenchyma, the mammary capsule was separated via blunt dissection and the mammary parenchyma was excised; the incision depth was 3.5–4 cm to ensure that the lactating tissue was obtained. The mammary parenchyma was subsequently washed with diethyl pyro carbonate-treated water. Approximately 1 g of tissue was cut into small pieces and immediately stored in liquid nitrogen for total RNA extraction.

### 2.4. Microarray

#### 2.4.1. Hybridization of DNA Microarray and RNA Extraction

The Agilent Bovine (V2) Gene Expression Microarray (Catalog Code: G2519F-023647; 4 × 44 K format) was used, with each array containing probes interrogating about 17,764 Entrez Gene ID in the array. A total of 36 microarrays were used for hybridization. Total RNA was extracted from each yak mammary tissue (50–100 mg) using Trizol reagent (Invitrogen, Karlsruhe, Germany) and purified according to the manufacturer’s protocol. The purity and concentration of RNA were determined by OD260/280 using a spectrophotometer (NanoDrop ND-1000). RNA integrity was determined by capillary electrophoresis using the RNA 6000 Nano Lab-on-a-Chip kit and the Bioanalyzer 2100 (Agilent Technologies, Santa Clara, CA, USA). RNA integrity number was >6 for all samples.

#### 2.4.2. RNA Amplification

Eberwine’s linear RNA amplification was performed by double-stranded cDNA containing the T7 RNA polymerase promoter sequence synthesis from 100 ng total RNA using the CbcScript reverse transcriptase with cDNA synthesis system according to the manufacturer’s protocol (Capitalbio) with T7 Oligo (dT). After completion of double-stranded cDNA (dsDNA) synthesis using DNA polymerase and RNase H, the dsDNA products were purified using a PCR NucleoSpin Extract II Kit (Macherey-Nagel, Düren, Germany) and eluted with 30 μL elution buffer. The eluted dsDNA products were vacuum-evaporated to 16 μL and subjected to 40 μL in vitro transcription reactions at 37 °C for 14 h using a T7 Enzyme Mix. The amplified cRNA was purified using the RNA Clean-up Kit (Macherey-Nagel, Düren, Germany).

#### 2.4.3. Labelling and Hybridization

cDNA was labeled with a fluorescent dye (Cy3-dCTP) produced by a Klenow enzyme labeling strategy after reverse transcription (Liu Zhihao et al., 2020). Briefly, 2 μg cRNA was mixed with 4 μg random nanomers, denatured at 65 °C for 5 min, and cooled on ice. Then, 5 μL of 4×first-strand buffer, 2 μL of 0.1 M DTT, and 1.5 μL CbcScript Ⅱ reverse transcriptase were added. The mixture was incubated at 25 °C for 10 min, then at 37 °C for 90 min. The cDNA product was purified using a PCR NucleoSpin Extract II Kit (Macherey-Nagel, Düren, Germany) and vacuum-evaporated to 14 μL. The cDNA was mixed with 4 μg random nanomer, heated to 95 °C for 3 min, and snap-cooled on ice for 5 min. Then, 5 μL Klenow buffer, dNTP, and Cy3-dCTP (GE Healthcare) were added to a final concentration of 240 μM dATP, 240 μM dGTP, 240 μM dTTP, 120 μM dCTP, and 40 μM Cy-dCTP, followed by addition of 1.2 μL of the Klenow enzyme. The reaction was performed at 37 °C for 90 min. Labeled cDNA was purified with a PCR NucleoSpin Extract II Kit (Macherey-Nagel, Düren, Germany) and resuspended in elution buffer. Labeled controls and test samples labeled with Cy3-dCTP were dissolved in 80 μL hybridization solution containing 3 × SSC, 0.2% SDS, 5 × Denhardt’s solution, and 25% formamide. DNA in hybridization solution was denatured at 95 °C for 3 min prior to loading onto a microarray. Array hybridization was performed in an Agilent hybridization oven overnight at a rotation speed of 20 rpm at 42 °C and washed with two consecutive solutions (0.2% SDS, 2 × SSC at 42 °C for 5 min, and 0.2 × SSC for 5 min at room temperature). The array data were analyzed for data summarization, normalization, and quality control using GeneSpring software V12 (Agilent). Transcriptome data were submitted to the NCBI public repository (accession number: GSE201438).

### 2.5. Validation of Microarray Data with RTqPCR

Validation of the microarray data was performed by comparing microarray data with the RTqPCR data of twelve transcripts, including *RPS15*, *RPS23*, *UXT*, *TP53*, *FARP1*, *SLC1A5*, *LPL*, *FABP3*, *SCD1*, *AGPAT6*, *CSN3*, and *BDH1* from the same yak mammary tissues that were previously analyzed [17,23,24].

### 2.6. Bioinformatics Analyses

The Dynamic Impact Approach (DIA) [25] was used for bioinformatics analysis. The DIA was specifically developed for analysis of dynamic transcriptomic changes, and was validated using bovine mammary tissue data throughout the whole lactation [3,25], a very similar experimental design to the present study. The DIA provides two outputs: the “Impact”, which calculates the impact of a condition (e.g., phase of lactation) on specific biological terms (e.g., the Kyoto Encyclopedia of Genes and Genomes (KEGG) pathway or Gene Ontology (GO)) using the proportion of DEG, the log_2_ expression ratio, and the −log_10_
*p*-value of the comparison. By calculating the difference of the Impact between genes that induce the term and genes that inhibit the term, the tool provides the “Direction of Impact”. Specific details of the tool are available elsewhere [25]. The DIA was used to analyze KEGG pathways and chromosomes.

Enrichment analysis of DEG was performed using Database for Annotation, Visualization, and Integrated Discovery (DAVID) v6.8 [26] and Ingenuity Pathway Analysis (IPA). Data were run in both tools using the default settings. In DAVID, three analyses were performed for each comparison: all DEG, up-regulated DEG, and down-regulated DEG. The last two analyses were used to determine whether the enriched KEGG and GO process terms were induced or inhibited. Results were downloaded using Chart Analysis. For IPA, the *p*-value and Z-score of each canonical pathway were downloaded. The Z-score in IPA infers activation states of each pathway (positive if increased and negative if decreased). For both tools, the annotated array was used as background.

### 2.7. Statistical Analyses

Data were analyzed by JMP genomics (version 7.0; SAS) using the default ANOVA model with time as the main effect and yak (n = 4) as a random effect. The overall effect of time was corrected by Benjamini–Hochberg False Discovery Rate (FDR) [27]. Differentially expressed genes (DEG) were determined when the overall time effect was with the FDR being ≤0.05 and the *p*-value between each comparison was ≤0.05.

## 3. Results

### 3.1. Overall Transcriptome Expression Profile in Yak Mammary Tissue

The complete transcriptome data after statistical analysis are available in Appendix A. Microarray data for a few selected genes through the lactation in yak were validated using RTqPCR. The data indicated a good concurrence between the data obtained from the two approaches (Appendix A).

The curve of lactation is reported in Figure 1A, with peak milk yield at 120 d and a rapid decline until 240 d. The statistical analysis of the mammary tissue transcriptome in yaks during the whole lactation identified 6256 DEG (Figure 1B). The number of down-regulated DEG was larger than that of up-regulated DEG throughout the entire lactation, although the up-regulated DEG had overall greater fold change (Figure 1B). Compared to −30 d, the number of DEG at each time point increased steadily until 30 d and remained high until 180 d, when a sharp drop in the number of DEG was detected, with almost no DEG in the 240 vs. −30 d comparison (Figure 1C). When compared with the previous time point, the largest increase in the number of DEG was detected between 240 d vs. 180 d, followed by the comparison between 1 d and −15 d (Figure 1D). The number of up- and down-regulated DEG vs. −30 d was similar among each time point vs. −30 d, with a relatively lower number of up- vs. down-regulated DEG between 30 d and 60 d. When considering the comparison between each consecutive time point, a lower and higher number of up- vs. down-regulated was observed in DEG between 30 d and 15 d and between 120 d and 60 d, respectively (Figure 1D).

### 3.2. Impact of Stage of Lactation on Chromosomes

The results of *Bos taurus* autosome (BTA) analysis by DIA are available in Appendix A. The top five most impacted chromosomes in mammary tissue during lactation in yaks were BTA6, BTA29, BTA23, BTA28, and BTA8 (Figure 2). Gene expression from most of the chromosomes was inhibited (*p* < 0.05) during lactation, except BTA6, BTA9, and BTA28, which were obviously activated (*p* < 0.05) during the period from day 1 to 60. In contrast, gene expressions from BTA3, BTA4, BTA14, and BTA20 were the most inhibited (*p* < 0.05) during lactation (Figure 2, Information and location in each chromosome of genes, is available in Appendix A).

### 3.3. Functional Analysis of DEG Revealed by DIA

The Impact and Direction of Impact of the categories of the KEGG pathways as analyzed using DIA are available in Figure 3. The same metrics for each pathway are available in Appendix A.

Metabolic-related pathways were activated overall during maximal milk yield increase, i.e., between 1 d and 60 d compared to −30 d, as well as at the onset of lactation, i.e., 1 d vs. −15 d. Among the most impacted sub-categories of pathways in the ‘Metabolism’ category were the sub-categories ‘Lipid Metabolism’, ‘Amino Acid Metabolism’, and ‘Metabolism of Cofactors and Vitamins’. The former two pathways were inhibited from 120 d on vs. −30 d, while the latter remained activated until the end of lactation. Among ‘Lipid Metabolism’ pathways, the most impacted and activated were ‘Synthesis and degradation of ketone bodies’, ‘Steroid biosynthesis’, ‘Glycerolipid metabolism’, ‘Glycerophospholipid metabolism’, ‘Arachidonic acid metabolism’, and ‘Linoleic acid metabolism’. The first two pathways were among the most impacted (see the ‘Pathway sort’ tab in Appendix A).

Among ‘Amino Acid Metabolism’, pathways associated with biosynthesis of Arg, Ile, Leu, Phe, Pro, Trp, Tyr, and Val were overall induced during first 60 days of lactation, while pathways associated with metabolism (i.e., degradation) of His, Lys, Trp, Tyr, Val, and taurine were overall inhibited during lactation. The ‘Phenylalanine, tyrosine and tryptophan biosynthesis’ pathway was the second most-impacted and induced pathway (see the ‘Pathway sort’ tab in Appendix A), while ‘Cysteine and methionine metabolism’ was induced only at the onset of lactation and between 180 d and 240 d, and was inhibited from 15 d to 180 d vs. −30 d. All pathways related to amino acid metabolism were induced at onset of lactation and further induced from 180 d to 240 d vs. −30 d except for taurine, which was inhibited at the onset of lactation. The ‘Metabolism of Cofactors and Vitamins’ category of pathway was highly impacted and activated during early and mid-lactation (i.e., in all time points vs. −30 d up to 180 d) and at the onset of lactation, and was inhibited between 180 d and 240 d vs. −30 d. Particularly affected in this category of pathways were ‘Riboflavin metabolism’, ‘Nicotinate and nicotinamide metabolism’, ‘Pantothenate and CoA biosynthesis‘, ‘Folate biosynthesis’, and ‘Retinol metabolism’.

‘Glycan Biosynthesis and Metabolism’ was impacted, and was slightly induced overall (Figure 3). The most impacted pathways in this sub-category of KEGG pathways were ‘O-Glycan biosynthesis’ and ‘N-Glycan biosynthesis’, which were strongly induced during lactation while being inhibited from 180 d to 240 d vs. −30 d, and ‘Glycosylphosphatidylinositol(GPI)-anchor biosynthesis’, which was overall induced during lactation, while ‘Glycosaminoglycan biosynthesis—heparan sulfate’ was strongly inhibited during the same time frame.

All the pathways in the ‘Replication and Repair’ sub-category of pathways were highly inhibited starting two weeks prior parturition and during lactation, then induced between 240 d and 180 d. Similarly, the sub-category of pathways ‘Growth and Death’ was highly inhibited during lactation and induced at the end of lactation (i.e., 240 d vs. 180 d). ‘Cell cycle’ was the most impacted and inhibited among the pathways in the ‘Growth and Death’ sub-category of pathways.

The sub-category of ‘Immune System’ and ‘Endocrine System’ pathways were highly impacted, with the former being mostly inhibited during and while the latter activated in the early stages of lactation and inhibited during the rest of lactation, with activation between 180 d and 240 d. The most impacted pathways in this sub-category of pathway were ‘Immune System’, ‘Chemokine signaling pathway’, ‘Toll-like receptor signaling pathway’, and ‘Complement and coagulation cascades’, which were overall activated, especially during early lactation; on the other hand, ‘Antigen processing and presentation’ was inhibited during the whole lactation. Among the ‘Endocrine System’ sub-category of pathways, the ‘PPAR signaling pathway’ was the most impacted, being induced during the whole lactation and inhibited between 180 d and 240 d. In the same sub-category of pathways, the ‘Renin-angiotensin system’ and ‘Progesterone-mediated oocyte maturation’ were highly impacted and inhibited during lactation, while being induced between 180 d and 240 d.

### 3.4. Enrichment Analysis

Complete results from DAVID are available in Appendix A. The KEGG pathways and Gene Ontology Biological Process terms enriched with a *p*-value < 0.01 are shown in Figure 4. The most enriched GO terms were associated with the cell cycle and proliferation, and were all enriched in down-regulated DEG in each time point during lactation vs. −30 d while being enriched in up-regulated DEG in the comparison of 240 d vs. 180 d. Enriched in down-regulated DEG during lactation were the GO terms associated with response to radiation (UV and gamma) and immune response. Among KEGG pathways, the most enriched among down-regulated DEG during lactation, though up-regulated between 240 d and 180 d, were pathways associated with the cell cycle. Others pathways similarly enriched were those associated with metabolism, with Cys and Met metabolism and pyrimidine metabolism enriched in up-regulated DEG at the onset of lactation and at the end of lactation; additionally, several time points during early and late lactation were enriched in down-regulated DEG. Steroid biosynthesis was enriched in up-regulated DEG between 1 and 60 day in lactation vs. −30 d. Protein processing in ER was increased at the onset of lactation and at 30 d compared to −30 d.

The complete results of the IPA analysis are available in Appendix A. The 25 most enriched (*p* < 0.05) canonical pathways are shown in Figure 5. Pathways that were estimated to be down-regulated by DEG during lactation (though up-regulated between 240 d and 180 d) were mostly involved with the cell cycle (including ATM Signaling, Estrogen-mediated S-phase Entry, and Role of CHK Proteins in Cell Cycle Checkpoint Control). Pathways estimated to be up-regulated were ‘Aryl Hydrocarbon Receptor Signaling’, ‘iNOS signaling’, and ‘OX40 Signaling Pathway’. Other pathways associated with immune response and the cell cycle (e.g., ‘Allograft Rejection Signaling’, ‘Antigen Presentation Pathway’, and ‘Cell Cycle Control of Chromosomal Replication’) were enriched.

## 4. Discussion

In the present study, a transcriptome profile of yak mammary tissue during the whole lactation was performed and compared with the same analysis performed in dairy cow [3].

### 4.1. Impact of Lactation on Yak Chromosome

Analysis of DEG affected by lactation and associated with chromosomes is critical in order to confirm chromosomal regions undergoing transcription changes during lactation. This can help to provide information about quantitative loci (QTL), which can benefit genetic selection to increase milk production and quality in yaks. For example, in the present study BTA20 was remarkably impacted during lactation, and harbors *GHR* and *PRLR*, both of which influence protein and milk yield [28,29,30]. As for dairy cows [25], in the present work on yaks the analysis of the transcriptome during lactation in mammary tissue revealed BTA6 as one of the most impacted and activated chromosomes, confirming it to be highly associated with QTL for milk production [31,32]. In dairy cows, casein coding genes are present in the BTA6 clustered in a small region together with statherin, all of which had a strong induction of transcription during the lactation cycle in yak mammary tissue (Appendix A) as well as in bovine mammary tissue [25]. In addition to caseins, BTA6 has a plethora of other genes that were up-regulated during lactation (Appendix A; e.g., *ABCG2* and *SPP1*), with a potential strong effect on milk yield. For instance, *ABCG2* plays a potential role in affecting the amount of water drawn into milk in vivo, thereby influencing milk volume [33,34].

Although most of the genes in BTA14 were downregulated from 1 d to 180 d during lactation in our study, several QTL exist in BTA14 that are critical for milk fat yield [28,35,36]. For instance, protein coded by DGAT1 influences milk fat synthesis [37]. The DGAT1 gene was up-regulated two-fold during lactation (from 1–180 d) in our study (Appendix A), which is consistent observations in bovines [38].

### 4.2. The Onset and End of Lactation in Yak Mammary Gland

In mammals, the transition from pregnancy to lactation is characterized by metabolic adaptation of major organs (e.g., mammary, liver), enabling animals to adjust to the need to synthesize milk for the newborns [39]. In our study, the number of DEG indicates that transcriptional regulation in yak mammary gland is more intense at the onset and end of lactation. These data are very similar to the bovine mammary transcriptomic data [3], supporting a strong role of the transcriptome in the adaptation of the mammary gland to lactation. Similar to prior data in bovine [3], the onset of lactation is characterized by pathways associated with metabolism, especially lipid and amino acid metabolism. Pathways related to lipid metabolism revealed an increase in synthesis of triglycerides, which is very similar to observations in the bovine mammary transcriptome [3].

The DEG in the comparison between 180 and 240 d in yak mammary tissue were highly associated with induction of ‘cell growth and death’, ‘immune system response’, ‘replication and repair’, and ‘amino acid metabolism’. The comparison between 240 d vs. 180 d corresponded to the time between end of lactation and dry period for yak. Thus, the comparison would be associated with mammary gland involution, in which more genes associated with ‘cell death’ and ‘immune response’ were induced. These data are consistent with the transcriptome of the mammary tissue in dairy cows [3], as well as with the typical large increase in mammary remodeling during involution [40,41].

The large impact and activation of the ‘Metabolism of Cofactors and Vitamins’ sub-category of pathways in the mammary tissue of yaks differed from the inhibition of the same pathway observed in bovine mammary tissue [3]. High impact and induction were observed in pathways associated with mitochondrial respiration in yak mammary tissue during lactation (e.g., riboflavin, Pantothenate and CoA) [42]. Changes in mitochondrial abundance and function have been observed in people living in high vs. low altitude [43], and the same difference is present between yak and cattle [44]. GO Cellular Component terms related to mitochondrial structure and function were enriched in up-regulated DEG from 1 to 30 d, and enriched in down-regulated DEG afterwards, supporting a role of mitochondria during lactation. However, the ‘Oxidative phosphorylation’ pathway was barely impacted in the DIA and was not enriched in DAVID or IPA.

The high impact on the ‘Folate biosynthesis’ pathway might indicate a role of this pathway in controlling the transcriptome. In folate metabolism, the 5,10-methylenetetrahydrofolate reductase (*MTHFR*) plays a key role in irreversibly converting 5,10-methylenetetrahydrofolate to 5-methylenetetra-hydrofolate, which is the main circulating form of folate. Mutations in the MTHFR gene, which is part of the ‘Metabolism of Cofactors and Vitamins’ pathway, is associated with changes in milk production in dairy goats [45]. The *MTHFR* transcript was upregulated during lactation in the present study (Appendix A). Interestingly, this gene is associated with a genetic adaptation to high radiation in Tibetan people [46].

In yak mammary tissue, ‘Cell growth and death’ was drastically induced at day 180 d compared with 240 d, and the fact that the lactation period of yak is around 180 days suggested an association with mammary gland involution, i.e., more genes associated with cell death were induced.

### 4.3. Lipid Metabolism Is Induced in Mammary Tissue of Yak during the Lactation Cycle

Milk solids are composed of lactose, fat, and protein. Lipid metabolism was induced in mammary tissue during lactation in our study, similar to bovines [3], humans [4], and rodents [47]. Among the most induced lipid metabolism pathway in the yak mammary tissue during lactation was the ‘Synthesis and degradation of ketone bodies’. The expression of 3-hydroxybutyrate dehydrogenase type I (*BDH1*) was up-regulated between five and ninety-fold during the whole lactation compared to −30 d in our study (Appendix A). This gene is the first step in the utilization of ketone bodies for the synthesis of milk fat [48,49], suggesting an important contribution of ketone bodies to milk fat synthesis in yaks. Recent data from buffalo suggests a similar role in that species [50] as well as in rats [51].

Interestingly, as observed in dairy cows [3], ‘Steroid biosynthesis’ was revealed to be induced during lactation by DIA, IPA, and DAVID, indicating an important role of the mammary tissue in cholesterol synthesis. Milk contains cholesterol, and in bovines it is thought to be mostly derived from the liver [52], although evidence exists of cholesterol synthesis in the mammary gland [53]. While the importance of cholesterol synthesis by the mammary gland is unclear, the consistent increase in expression of cholesterol synthesis-related genes in mammary tissue of yak and cow, as observed in humans [4], strongly suggests an important role of cholesterol synthesis in milk production.

Increases in milk fat synthesis shown by the transcriptome data were supported by the induction of several pathways involved in triglyceride synthesis; very similar pathways are induced in the mammary tissue of dairy cows during lactation [3]. These data confirm the important role of the transcriptome in controlling milk fat synthesis [38]. Additional pathways that support increased lipid synthesis in yak mammary tissue during lactation include the induction of ‘Riboflavin metabolism’, which was largely caused by the 11-to-32-fold increase in transcription of *ENPP3* (ectonucleotide pyrophosphatase/phosphodiesterase 3) during lactation vs. −30 d (Appendix A). The function of the enzyme encoded by this gene is to synthesize the cofactors FAD and FMN using riboflavin [54], which is essential for the proper function of the pentose phosphate pathway (not induced in our data). This is the main pathway for the provision of NADPH for the synthesis of lipids [55].

Another important pathway associated with lipid metabolism was the ‘Pantothenate and CoA biosynthesis’. This pathway provides Coenzyme A for the activation of fatty acids prior to their utilization in the milk fat synthesis [56]. ‘Linoleic acid metabolism’ was strongly induced in yak mammary tissue, though it is not affected in dairy cows during lactation [3]. Yak milk is characterized by high concentration of conjugated linoleic acid, which is derived from the biohydrogenation of dietary linoleic acid in the rumen [57,58].

The strong induction of the ‘PPAR signaling pathway’ during lactation in yak mammary tissue reflects the same induction observed in dairy cows [3,25,59]. These data confirm the importance of Peroxisome Proliferator-activated Receptors in regulation of milk synthesis; the role of PPARγ appears to be especially important in controlling milk fat synthesis, as has been previously argued [7,38,60]. A study of long non-coding RNA in yak mammary tissue during the dry period and lactation additionally underscores an important role of PPAR in lactation [18].

### 4.4. Amino Acid Metabolism

Pathways related to amino acids indicate a strong role of the mammary gland in synthesis and utilization of amino acids, similar to bovine transcriptome data [3,20]. The importance of amino acid synthesis during lactation has been revealed by a recent study on long non-coding RNA in yak mammary tissue [18]. The pattern of ‘Valine, leucine and isoleucine degradation’ and the ‘Cysteine and methionine metabolism’ pathway were very similar between yak and bovine. However, the pattern of other amino acid-related pathways differed in yak compared to bovine. While in bovine these pathways were consistently activated during the whole lactation, in yak the pathways were activated early on during lactation and inhibited afterwards.

As for bovine, these data indicate an important role of amino acid metabolism in the mammary gland, including the sparing of Met for the synthesis of milk proteins, as previously argued [3]. Transcriptome data, both microarray (as in the present manuscript) and RTqPCR [17], support an increase in expression of genes related to the regulation of milk protein synthesis, similar to the data for dairy cows [3]. RTqPCR data [17] suggest an important role of amino acid transport for milk protein synthesis. Rather than an increase in milk protein synthesis machinery, the present data suggest an increase in the metabolism of AA, providing additional support for an important role for the mammary tissue in handling amino acids for milk protein synthesis.

In bovines as well as in other species, the expression of ribosomes mostly decreases during lactation, while transcription of genes coding for caseins and other typical milk proteins strongly increases, suggesting specialization of the mammary gland to synthesize milk proteins and not proteins for ancillary functions in the mammary gland [20]. In yak, the transcription of ribosomes was not strongly impacted; however, for other species [20], RNA degradation and protein degradation (i.e., proteasome) are inhibited and ER processing of proteins is induced during lactation, supporting an increase in synthesis of proteins that are secreted.

### 4.5. Glycan Biosynthesis and Metabolism

Bovine milk N-glycome during early lactation is highly affected, with the alteration being largely due to the changes in N-glycosylation of IgG in colostrum [61]. Nearly forty oligosaccharides were detected in dairy cow milk, most of which stimulate the growth of beneficial bacteria and inhibit pathogen binding to epithelial cell surfaces in the intestine [62]. Among these, glycosphingolipids have attracted the most attention because of their probiotic roles [62]. In mammary tissue of dairy cows, ‘Glycosylphosphatidylinositol(GPI)-anchor biosynthesis’ is among the most impacted and induced pathways [3]. Although the role of this pathway in lactating mammary tissue remains unknown, it is of interest that in yaks this pathway appears to be associated with milk synthesis, prompting further research to understand its role.

N-Linked glycans are attached in the endoplasmic reticulum, and O-linked glycans are assembled in the Golgi apparatus to produce modifications to proteins, as in glycoproteins and proteoglycans [63,64]. The strong induction of O-glycan and N-glycan biosynthesis in mammary tissue during lactation in yak strongly suggests enrichment of glycoproteins and proteoglycans in yak milk.

### 4.6. Cell Cycle

Cell cycle-associated pathways and GO terms were strongly inhibited according to DIA, and were the most enriched according to DAVID and IPA during lactation, though activation between 180 d and 240 d strongly suggested the cell cycle to be a major transcriptomic adaptation in lactating mammary tissue in yaks. Cell cycle and apoptosis were found to be important in mammary tissue of dairy cows [3]. As for dairy cows, the data clearly indicate that the mammary gland of the yak does not change its morphology/structure or experiences minimal changes in terms of cell profile during lactation (or even two weeks prior lactation). In contrast, strong morphological changes take place during the end of lactation (i.e., 240 d vs. 180 d).

The microarray data indicated an important role of the ‘p53 signaling pathway’ in controlling the cell cycle in the yak mammary gland. A previous study with the human mammary MCF10A cell line showed that elevated p53 protein levels can arrest the cell cycle and initiate apoptosis [65,66], which further supports the stasis of cell proliferation suggested above. The increase in apoptosis after peak milk yield supports a likely gradual decrease in number of mammary epithelial cells after peak lactation, which might account for the decline in milk production with advancing lactation, as observed in dairy cows [67].

### 4.7. Immune Response System

Impact on functions related to the immune system in mammary tissue would be expected due to the evolutionary origin of the mammary gland and its known roles in the immune system [35]. The pattern of pathways associated with immune system in yak mammary tissue observed in the present work were very similar to that observed in dairy cows [3].

The ‘Complement and coagulation cascades’ pathway belongs to the innate immune system. The activation of this pathway during lactation could be taken to mean an increase in the ability of ‘complement’ to kill pathogens by antibodies [68]. The induction of the ‘Toll-like receptor signaling pathway’ during lactation implies an increasing ability for mammary tissue to recognize invading microbes, especially gram-negative bacteria that present lipopolysaccharide [69].

Similar to dairy cows, the overall induction of pathways associated with innate immune response, such as ‘Complement and coagulation cascades’ and ‘Toll-like receptor signaling’, along with strong inhibition of ‘Antigen processing and presentation’, appears to support previous conclusions [3] that lactating mammary tissue experiences an increase of amount/activity of immune cells with a concomitant decrease in the hyper-activation of the immune system. Considering the impact of mastitis in the dairy industry, this consistent finding between the two Bos species is important.

### 4.8. Adaptation to Low Oxygen and High Radiation Environment

The metabolism of the yak is adapted to the low oxygen content on the Tibetan plateau [70]. Among pathways potentially associated with the response to low levels of oxygen, the ‘VEGF signaling pathway’ was induced in DIA in the present study, and was not enriched in DAVID or IPA. Vascular endothelial growth factor (VEGF) acts as a signal protein to stimulate vasculogenesis and angiogenesis, restoring oxygen supply to tissues when blood circulation is inadequate. Transcription regulation of VEGF and the VEGF receptor plays a role in rodent mammary gland function during pregnancy, lactation, and involution [71], while its role in the dairy cow mammary gland during lactation appears to be minor [3]. Inactivation of VEGF in the mammary gland epithelium severely compromises mammary gland development and function [72]. Unlike dairy cattle, yaks must adapt to an environment with a low oxygen concentration. Therefore, induction of the ‘VEGF signaling pathway’ could be a potential adaptive mechanism in the yak mammary gland.

As there is high solar radiation at the high altitude of the Tibetan plateau [73], the body has to adapt to this amount of radiation, as observed in humans [46] and, although not studied yet, likely in yak as well [74]. Because yaks must lactate under high solar radiation, the enrichment in down-regulated DEG of pathways associated with response to radiation is somewhat puzzling [75]. It is possible that the response to radiation is an important function in all yak tissues; the functional adaptation of the mammary gland to produce milk might involve the decrease of other biological functions of mammary tissue that are not teleologically ordered towards milk synthesis.

### 4.9. Limitations

This study presents several limitations. The use of microarray instead of RNAseq limits our ability to detect more transcripts, and microarray has lower sensitivity and reliability compared to RNAseq. Another potential limitation in our approach is that in mammary tissue the change in transcription of few highly expressed genes, such as caseins and lactalbumin, can “dilute” the level of other transcripts, generating an artificial down-regulation of transcripts, as previously demonstrated for RTqPCR (Bionaz and Loor, 2007). This would of course affect the interpretation of our findings. The dilution effect mentioned above can be very problematic when using quantitative techniques such as RTqPCR or RNAseq, though it might be not as problematic when using microarray. This is somewhat demonstrated by work done on pig mammary tissue during the lactation cycle of several transcripts, which was not affected by the day relative to parturition as potential internal control genes (Tramontana et al., 2008). Those genes were all “down-regulated” in the RTqPCR data prior to normalization. In order to evaluate whether a dilution effect was present in our microarray data, we evaluated the pattern of several transcripts previously used as references genes in the two above-cited papers. Fifty percent of the transcripts considered had a significant down-regulation that could be due to a dilution effect. We have also evaluated the three best reference genes for RTqPCR analysis of the same samples used in the present experiment (Jiang et al., 2016), and found that two transcripts were significantly affected through lactation, with *MRPS15* being down-regulated (i.e., an apparent dilution effect), *UXT* having an increase through lactation, and *RPS23* not being significantly affected (Appendix A). Overall, we cannot exclude the possibility that certain down-regulated pathways may be a consequence of an artificial “dilution” of their transcripts; however, our data do not provide evidence that any dilution was apparent in our microarray data. In any case, this is an issue with normalization of RTqPCR vs. RNAseq or microarray that prevents full comparison between these techniques, which is an issue that lies outside the purpose of the present manuscript.

## 5. Summary and Conclusions

The transcriptome profile of yak mammary tissue during lactation was displayed by sampling nine time points. DIA was used to identify lactation-related genes on yak chromosomes, laying a foundation for QTL mapping of the chromosomes. KEGG analysis indicated that transcriptional regulation in the yak mammary gland is more intense at the onset and end of lactation. Functional analysis underscored the importance of induction of lipid and amino acid metabolism, glycan biosynthesis, and PPAR signaling along with inhibition of the cell cycle, response to radiation, and immune response during lactation. The transcriptome adaptation of mammary tissue to lactation in yaks appears to be very similar to that observed in dairy cows, though with several unique characteristics.

## Figures and Tables

**Figure 1 animals-13-01710-f001:**
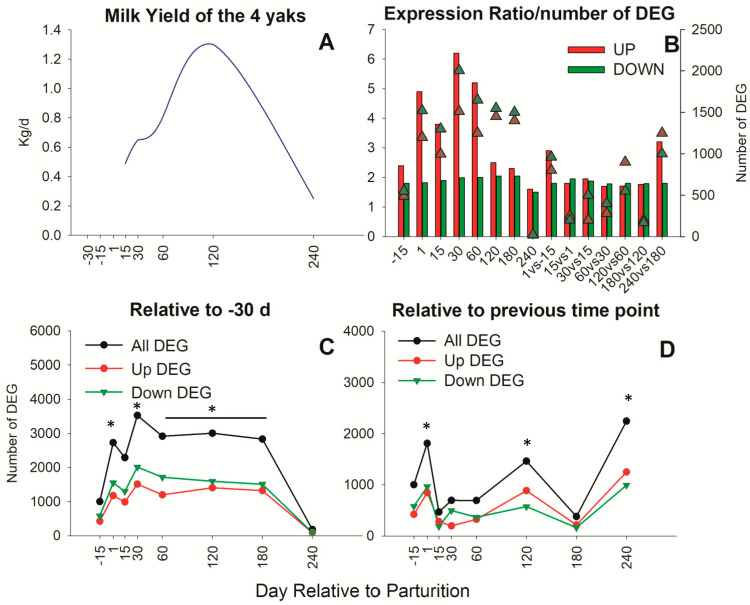
Milk production and transcripts differentially affected during lactation in mammary tissue of yaks. (**A**) Milk yield during lactation in yak. (**B**) Expression ratio (triangles) and number of up-regulated (red bars) and down-regulated (green bars) differentially expressed genes (DEG) between comparison at each time point vs. −30 d and comparison between consecutive time points. (**C**) Number of DEG at each time point relative to −30 d. (**D**) Number of DEG between consecutive time points. The asterisk (*) indicates significantly different.

**Figure 2 animals-13-01710-f002:**
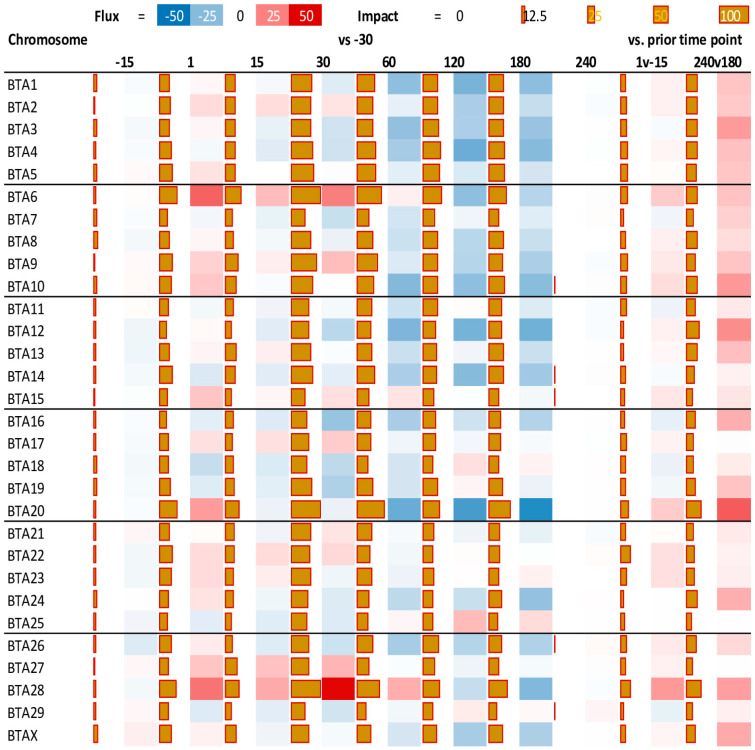
Impact of stage of lactation on chromosomes in yak mammary tissue. Dynamic Impact Approach results of differentially expressed genes in each time point vs. −30 d relative to parturition or relative to prior time point on chromosomes. Brown horizontal bars denote the impact and the square on their right denotes the direction of the impact (red = activation; blue = inhibition).

**Figure 3 animals-13-01710-f003:**
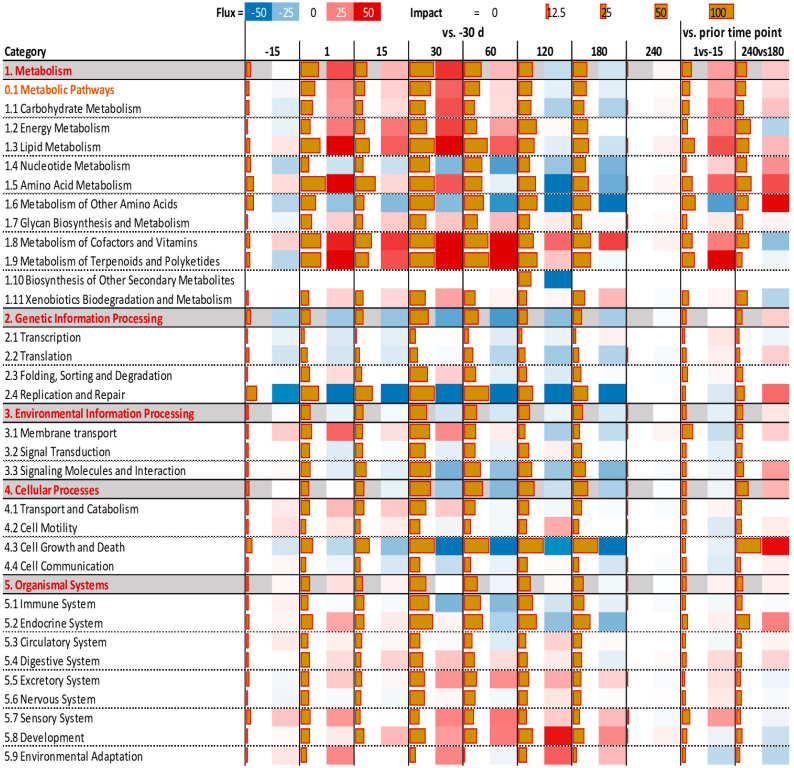
Impact of stage of lactation on KEGG pathways in yak mammary tissue. Dynamic Impact Approach results of differentially expressed genes in each time point vs. −30 d relative to parturition or relative to prior time point on main categories and sub-categories of KEGG pathways. Brown horizontal bars denote the impact and the square on their right denote the direction of the impact (red = activation; blue = inhibition).

**Figure 4 animals-13-01710-f004:**
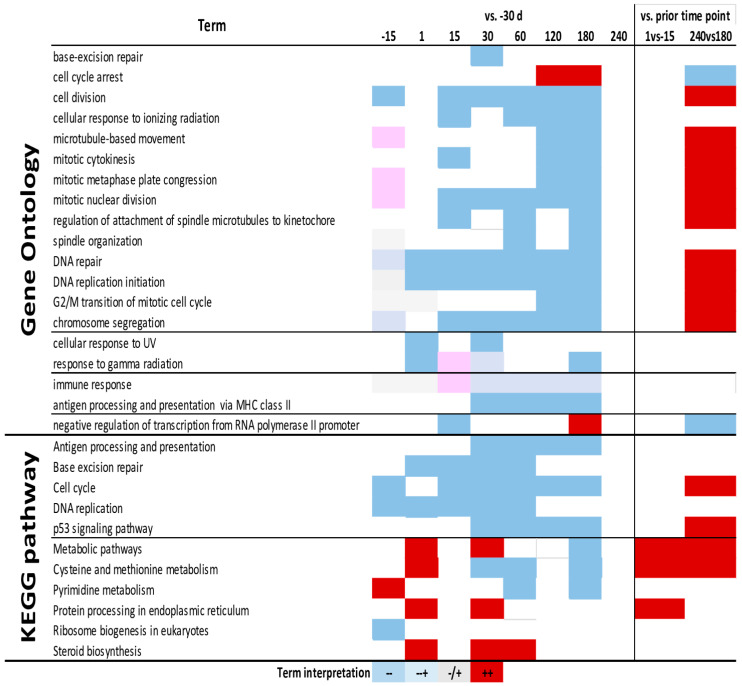
Gene Ontology and KEGG pathway terms enriched during lactation in yak mammary tissue. Most enriched (*p*-value < 0.01) Gene Ontology Biological Processes and KEGG pathways of differentially expressed genes affected during the lactation cycle in yak, as revealed by DAVID analysis. Estimated induced and inhibited terms are highlighted in red and blue, respectively (see Materials and Methods section for details).

**Figure 5 animals-13-01710-f005:**
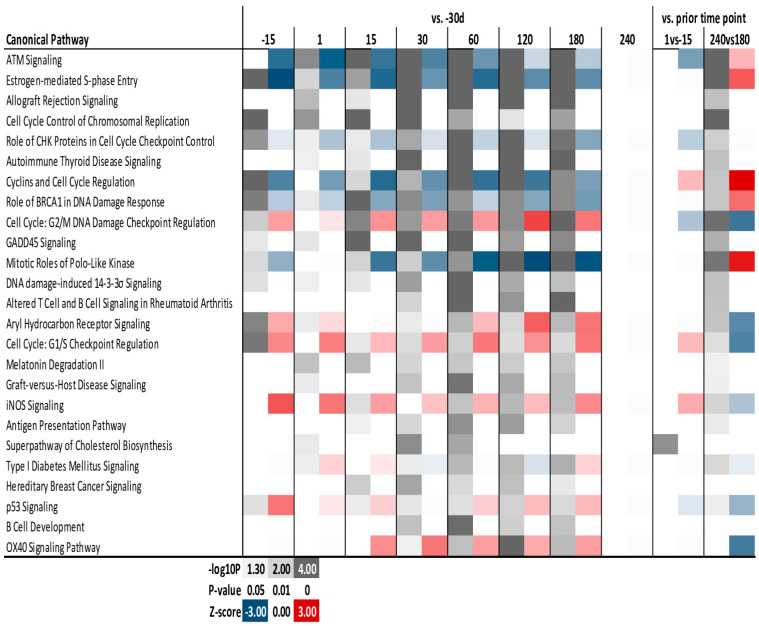
Canonical pathways in IPA affected by the lactation cycle in yak mammary tissue. The most enriched Canonical Pathways of differentially expressed genes affected during the lactation cycle in yak, as revealed by Ingenuity Pathway Analysis. Reported are the *p*-value of enrichment (grey shade) and estimation activation of inhibition of the pathway as indicated by Z-score (blue = inhibition and red = activation).

## Data Availability

Transcriptome data were available at the NCBI public repository (accession number: GSE201438).

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
