# Peer review of "Dynamic Profile of the Yak Mammary Transcriptome during the Lactation Cycle"

_animals, 2023, doi:10.3390/ani13101710_

Round 1
Reviewer 1 Report
This study entitled “Dynamic Profile of the Yak Mammary Transcriptome during the Lactation Cycle” explored the changing transcriptome of the mammary gland of yaks during parturition. The study could contribute to animal science by reporting the comprehensive transcriptome of the yak mammary gland during lactation. However, this paper needs to address several issues as the following suggestion
Major revision
Line 14-34; The number of animals used in this study was not written
Line 14-34: Objective of this study is not clear. State what is the objective of this study.
Linw 15: Were the samples taken from same individuals or from different individuals?
Line 16: State what “Baioinformatics tools” is.
Line 18: Why did this paper used both false discovery rate and p-value as threshold?
Line 20: State what “Bioinformatic analysis” is.
Line 23-31: It was not clear that the differences were observed in which comparison (which sampling day to which the other sampling day).
Line 43: The milk yield of dairy Holstein cows are over 3,000 kg. Therefore, this sentence is partially incorrect.
Line 46; Compared to which cows? Holstein?
Line 57-60: Most study use RNA-seq even if they use a number of samples.
Line 88: What time were two milkings done?
Line 99: Is this a proper noun?
Line 104-106: This should be the most important part of this study, but this study lacks enough information for the reader to reproduce the experiment.
Line 168: Why did this study not set the threshold for DEG analysis?
Line 180: Why did this study not set the threshold for Z-score?
Line 187: Adjusted p-value should be used here because this study compared multiple time points.
Figure 1B; What did the triangle and bar mean?
Line 332: What is the meaning of “not Z-score”?
Author Response
Reviewer 1
Comments and Suggestions for Authors
This study entitled “Dynamic Profile of the Yak Mammary Transcriptome during the Lactation Cycle” explored the changing transcriptome of the mammary gland of yaks during parturition. The study could contribute to animal science by reporting the comprehensive transcriptome of the yak mammary gland during lactation. However, this paper needs to address several issues as the following suggestion
Major revision
Line 14-34; The number of animals used in this study was not written
Response: I appreciate for your question and the number of animals had been added in the Abstract.
Line 14-34: Objective of this study is not clear. State what is the objective of this study.
Response: The abstract has been rewritten and an objective is now clearly stated.
Linw 15: Were the samples taken from same individuals or from different individuals?
Response: samples were taken from the same individuals.
Line 16: State what “Baioinformatics tools” is.
Response: We have now rewritten the abstract and we indicated that we used several bioinformatic tools. The reason for being general is the long names of each of the three tools used and citing all of them will make the abstract length behind the limited words required by the journal.
Line 18: Why did this paper used both false discovery rate and p-value as threshold?
Response: We used false discovery rate for the whole-time course and P-value difference between each comparison to identify differentially expressed genes . We have now clarified that in the abstract and the body of the manuscript.
Line 20: State what “Bioinformatic analysis” is.
Response: See above our comments for L16.
Line 23-31: It was not clear that the differences were observed in which comparison (which sampling day to which the other sampling day).
Response: This is now addressed
Line 43: The milk yield of dairy Holstein cows are over 3,000 kg. Therefore, this sentence is partially incorrect.
Response: Thanks for pointing this out! The sentence had been changed in the revised manuscript.
Line 46; Compared to which cows? Holstein?
Response: Yes, we performed the comparison with data from Holstein cows. We have added this in the revised sentence.
Line 57-60: Most study use RNA-seq even if they use a number of samples.
Response: Thank you for your comments. We used microarray instead of RNAseq for two reasons: this experiment was performed in 2013-2014, at that time the RNA-seq was very expensive and not affordable for all the samples we had; although RNA-seq has many advantages, many researchers continue to use microarray, especially in studies with large sample size. Microarray is more popular in clinical research because its data processing is fast and simpler than RNAseq.
Line 88: What time were two milkings done?
Response: We have added this information in the revised manuscript.
Line 99: Is this a proper noun?
Response: Changed into Dynamic Impact Approach (DIA).
Line 104-106: This should be the most important part of this study, but this study lacks enough information for the reader to reproduce the experiment.
Response: Thank you for pointing this out! We have added more details on the biopsy.
Line 168: Why did this study not set the threshold for DEG analysis?
Response: To select differentially expressed genes, we used both false discovery rate and p-value as threshold.
Line 180: Why did this study not set the threshold for Z-score?
Response: The Z-score in IPA mainly provides the direction of the effect (i.e., activated or inhibited). It could be used as threshold, but it is more appropriate to use the P-value as threshold and use the Z-score to aid in interpreting the effect.
Line 187: Adjusted p-value should be used here because this study compared multiple time points.
Response: We have addressed the multiple comparison by using the FDR of the overall time effect. Between comparison the use of a P-value, although more liberal, can still be valid. An additional multiple comparison correction can be done, but, usually a less stringent approach, such as Scheffe's Test. Our method is very similar to what was used in the bovine microarray data used for the comparison (see DOI: 10.1371/journal.pone.0033268).
Figure 1B; What did the triangle and bar mean?
Response: Thank for pointing this out. We have now clarify this in the caption of the Figure.
Line 332: What is the meaning of “not Z-score”?
Response: That was an error and now we have remove it.
Reviewer 2 Report
The manuscript describes the changes that occur in the mammary gland transcriptome during lactation. The authors used the currently unpopular method of expression microarrays. They observed numerous changes in gene expression levels over different point-times. Their direction corresponded to the generally accepted knowledge that secretory functions and biosynthesis of amino acids and fats increase during lactation.
Specific comments:
line 16 gene expression platform?
line 18 please select "genes" or "transcripts", which is more appropiate for microarray analysis?
line 91 How many Yaks were used for each analyzis?
line 97how was the sampling performed by biopsy?
line 221I am not sure if this approach is correct, there are plenty of genes on each chromosome. I do not see the rationale for calculation inhibition/activation ratio at the chromosome level. Perhaps it would be better to compare expression data with known QTLs
line 240 What do you mean by "lactation related pathways" - please specify these pathways
line 331please rewrite the sentence its confusing
line 334please rewrite the sentence its confusing
Supplementary material 1 Please provide time -point differences (At least between two-time points) obtained by qPCR and microarrays for the same gene.
I could not find supplemetary file 3 and 4
Author Response
Reviewer 2
Comments and Suggestions for Authors
The manuscript describes the changes that occur in the mammary gland transcriptome during lactation. The authors used the currently unpopular method of expression microarrays. They observed numerous changes in gene expression levels over different point-times. Their direction corresponded to the generally accepted knowledge that secretory functions and biosynthesis of amino acids and fats increase during lactation.
Specific comments:
line 16 gene expression platform?
Response: The abstract was rewritten, also to account for those unclear wordings.
line 18 please select "genes" or "transcripts", which is more appropiate for microarray analysis?
Response: The most appropriate is “transcript” when referring to the measured mRNA but “gene” when referring to the DNA locus where the mRNA is transcribed. We have now used the two consistent through the manuscript.
line 91 How many Yaks were used for each analyzis?
Response: Four yaks were used for transcriptome analysis.
line 97how was the sampling performed by biopsy?
Response: We have now added the details of the biopsy
line 221 I am not sure if this approach is correct, there are plenty of genes on each chromosome. I do not see the rationale for calculation inhibition/activation ratio at the chromosome level. Perhaps it would be better to compare expression data with known QTLs
Response: We agree that the comparison with known QTL is more useful. This is what we have done in the discussion. As the yak is a relatively novel investigate species using high-throughput transcriptomic analysis, it seems sensible to try to use the generated data to identify chromosome that appear to be more important during the transcriptomic adaptation to lactation.
line 240 What do you mean by "lactation related pathways" - please specify these pathways
Response: We agree that this was confusing. We have removed the sentence.
line 331please rewrite the sentence its confusing
Response: We agree that the sentence was awkward. We have removed it.
line 334please rewrite the sentence its confusing
Response: The sentence was rewritten.
Supplementary material 1 Please provide time -point differences (At least between two-time points) obtained by qPCR and microarrays for the same gene.
Response: We have provided a comparison between RTqPCR and microarray data in Suppl. Figure 1.
I could not find supplemetary file 3 and 4
Response: Sorry for the inconvenience. We have uploaded all the supplementary files in the revised manuscript.
Round 2
Reviewer 1 Report
This paper was improved after revision.
Author Response
Thank you so much for the efforts for reviewing this manuscript. Thank you!
Reviewer 2 Report
the manuscript still needs improvement. It contains a lot of mental shortcuts that make it difficult to fully understand, for example : "dynamic transcriptome',
moreover,
line 18 please rewirute the sentence
line 333 Impact of lactation on Yak chromosome?
Supplementary fig. 1 there is no standard errors and p-values on the graph
The interesing point of the manuscript is the identification of
| Role of BRCA1 in DNA Damage Response |
pathway, perhaps it could somehow explain a negative association between breast feeding and the occurance of breast cancer
Author Response
the manuscript still needs improvement. It contains a lot of mental shortcuts that make it difficult to fully understand, for example: "dynamic transcriptome',
response: The ‘dynamic transcriptome' had been changed into ‘transcriptome’ in the whole manuscript. The whole manuscript had been revised by the native speaker Massimo Bionaz again.
moreover,
line 18 please rewirute the sentence
response: Thank you so much for your question and I have rewrited the sentence.
line 333 Impact of lactation on Yak chromosome?
response: Thank you for your suggestion and I have made the change according to your suggestions.
Supplementary fig. 1 there is no standard errors and p-values on the graph
response: Thank you for your question and we only showed the expression pattern of QPCR and microarray data.
The interesting point of the manuscript is the identification of
|
Role of BRCA1 in DNA Damage Response |
pathway, perhaps it could somehow explain a negative association between breast feeding and the occurance of breast cancer
response: I appreciate for your suggestion and I totally agree with your idea. In next paper, we will explore the role of BRCA1 function in breast cancer.